# The Impact of COVID-19 Pandemic Containment Measures on Families and Children with Moderate and High-Functioning ASD (Autism Spectrum Disorder)

**Margarita Saliverou [1], Maria Georgiadi [2],\*, Dimitra Maria Tomprou [1,3], Nataly Loizidou-Ieridou [1] and Stefanos Plexousakis [1,2]**

[1] Psychology and Social Sciences, University of Frederick, P.O. Box 24729, Nicosia 1303, Cyprus; st017676@stud.frederick.ac.cy (M.S.); dledu.td@frederick.ac.cy or dmtomprou@primedu.uoa.gr (D.M.T.); pre.nl@frederick.ac.cy (N.L.-I.); dledu.ps@frederick.ac.cy or splexousakis@uoc.gr (S.P.)

[2] Applied Psychology Laboratory, Department of Psychology, University of Crete, 74100 Rethymno, Greece

[3] Department of Special Education and Psychology, Faculty of Primary Education, University of Athens, 10680 Athens, Greece

\* Correspondence: m.georgiadi@uoc.gr

**Abstract:** The present study focuses on the impact of SARS-CoV-2 (COVID-19) transmission prevention measures and, in particular, home confinement of families with children with autism spectrum disorder (ASD) in Greece. It is assumed that the implemented new measures during the pandemic constitute a profound change for children on the spectrum, considering that the core ASD symptoms include the persistence and adherence to routine and stability, a condition that also directly affects the children's parents. Semi-structured telephone interviews were conducted. Participants were 10 caregivers with a child diagnosed with ASD of medium or high functioning in Greece. The ages of the children range from 6.5 to 15 years old. The results of the thematic analysis revealed three main themes: (1) the educational framework, (2) the management of daily life, and (3) the construction of the new daily routine. These three themes represent the levels that have undergone a decisive transition, and the sub-themes recommend the areas, individual ways of dealing with this shift. So far, the impact of the pandemic mitigation measures cannot be described as generally positive or negative, as there have been advances and setbacks for children and families alike. Finally, governmental measures and technology-assisted teaching (distance learning) were considered necessary but not sufficient enough for full adaptation.

**Keywords:** autism spectrum disorder (ASD); autism; pandemic; COVID-19; families; special education

## 1. Introduction

The unprecedented social and political condition of the widespread of the SARS-CoV-19 virus has prompted governments worldwide to impose strict social distancing and home containment measures on a broad social scale [1], causing significant emotional and behavioral consequences on children with ASD. The main symptoms of autism spectrum disorders (ASD) include a particular sensitivity to change and persistence in specific routines and interests with a multitude/plethora of behavioral acts [2], a condition that, along with comorbidity, produces a challenge in the context of the COVID-19 pandemic. The pandemic presents even a more difficult challenge for people with autism due to high levels of anxiety, sensory issues (for example, they may face difficulty wearing masks), and their inability to adjust to the changes in their daily routine [3].

In the early literature, autism spectrum disorder (ASD) is recorded as a rare childhood disorder [4]. Over the last 30–40 years, however, scientific research has highlighted its lifelong nature and the increasing incidence rate [5]. The overall increase in the rate seems to be related to the intensified research interest in recent years and modifications in clinical features that have highlighted autism as more prevalent than other known and

widely diagnosed diseases, such as diabetes and heart disease [6]. The prevalence rate is determined to be 1–3% in children and adolescents, corresponding to 7.6 per 100,000 or more people worldwide [7–9]. The first large-scale survey in Greece recorded a prevalence rate of 1.15% and a 4.14:1 boys/girls ratio [10], which according to Looms et al. [11] is 3:1.

On 31 December 2019, China's World Health Organization (WHO) office recorded a potentially zoonotic pulmonary virus in Wuhan; an unknown pulmonary case—for which further information was needed—was also reported in ProMED [12]. An official international emergency condition will be declared by WHO on 30 January. On 11 March, the pandemic state was officially declared. By the end of February, the number of cases worldwide reached 85,493; specifically in Europe, 1119 incidents were recorded, while in Greece, only three vectors were listed. Although the possibility of a pandemic outbreak was not unknown, and there were warnings from the UN about the occurrence of a pandemic crisis that could not be supported by countries' health systems, and no action was taken until after the outbreak [13].

Like most countries, Greece was struggling to cope with the spread of the virus. On 26 February, «Patient Zero» was recorded [14], and the National Government declared an emergency state and started to enact legislative measures [15]. The generalized realization of the health systems' inability to support a pandemic crisis and the lack of a cure and vaccine led to the expanded implementation of health protection measures in the daily lives of citizens from the simple use of masks, social distancing, and regular hand and facial hygiene to the most effective measure: social isolation and quarantine [14]. The phases of the pandemic spread can be distinguished based on the number of cases and the measures imposed; the present study took place between the first wave/phase (P1) and the second one (P2). In the P1, Greece was listed among the countries with the least cases [16]. The expectation that the rise in temperature during the summer months would stop the virus was not confirmed, although a broader decrease in cases was observed [17]. This led to a second universal lockdown in November 2020 (P2) [18]. The vast majority of the Greek population complied with the restrictive measures in P1 and P2 [19–22]. The Greek population accepted the "stay at home" edict. The effects of these measures in the Greek population were social isolation, unemployment or changes to the employment status, and increased family conflicts [20,21].

Additionally, one of the main measures that all countries enforced, including Greece, was long-term school closure and the switch to distance education [23,24]. Especially P1 was characterized by the total lack of access to services, open care facilities, and after-school intervention programs for children with disabilities. The findings of research on the difficulty of achieving balance in the family of a child with autism due to the immense pressure that child's needs impose on the family [25]. In addition, the broader picture of special education in Greece is not ideal due to the s lack of personnel and suitable buildings with all the necessary equipment and accommodations for students with ASD [26,27]. The above factors create the necessity of investigating the level of impact and regression of rights in all areas of human activity for children with ASD, particularly in light of the persistence of the epidemic for years to come and the possibility of new pandemic waves emerging [28].

Increased stress levels and lower levels of well-being of parents of children with ASD were confirmed by a large number of studies [29] in comparison with parents of neurotypical children or children with other forms of disability [30–33]. Furthermore, it could often lead to burnout, depression, and psychological and physical breakdown, a condition that is exacerbated by deficits in services and general childcare [26,27,34].

The actual positive or negative effects of pandemic measures on the daily lives of children with autism and their families are still uncharted and unclear, as extensive international research has yet to be conducted [35,36]. A large number of preliminary research efforts have attempted to assess, evaluate, and predict the difficulties and progress that may arise based on the characteristics of the disorder [3]. It is clear that the health crisis has affected the socioeconomic sector as well as families and children with autism in terms

of the services they receive and autism-condition-related research by itself [37]. Previous epidemic crises (SARS, H1N1, MERS, EBOLA) cannot be compared to the current one, and this makes it difficult to systematically predict the consequences [38]. There are two main findings that also apply to the existing framework. Firstly, this condition is expected to affect vulnerable population groups profoundly [39,40]. Secondly, the demise of the pandemic is equivalent to the onset of the post-pandemic condition corresponding to the post-traumatic phase [37]. Furthermore, we must take into account that most syndromes co-occurring with autism are acting as risk factors for coronavirus patients [41]. In this perspective, the impact of house confinement and quarantine on children with autism and their families has emerged. Indeed, according to Brooks [39], it is recommended that mandatory isolation and home confinement should be time-limited and based on the individual decision of each person; otherwise, the effectiveness of the measures and the mental health of the individuals is compromised.

Strong assumptions emerge predicting increased anxiety and stress for carers and children who face a new daily routine with perhaps complete lack of access to support [35,42,43]. A multi-racial USA study confirms increased levels of anxiety, distress, and psychological burden for parents of ASD children as they experience even greater compression of personal space and time [44]. The main factors that influenced parents' emotional and mental health are children's age, gender, and severity of ASD, as confirmed in a quantitative study in Saudi Arabia [45]. Parents of children with autism in Turkey confirm the predictions about negative consequences of quarantine and report stronger pessimism compared to parents of neurotypical children [42]. At the same time, many parents were confronted with unemployment or remote working while facing new responsibilities and more challenges on a financial and psychological level, elements that combined with the possible limited home space is proven to burden the individual's capabilities and autonomy [46,47]. According to an online survey carried out in northern Italy, a very high percentage, 93.9%, of families found the quarantine to be a demanding period. In particular, they were confronted with increased difficulties in the level of management of the children's daily schedule, especially leisure time and structured activities. The children showed more pronounced and more frequent behavioral problems, while at the same time, there were emergency needs of receiving increased health support and assistance, particularly at home, to address the possibility of a disruptive quarantine [48–50].

Technology was a necessary and vital tool to assist the adjustment to the new condition. Countries were urgently invited to establish and embrace the world of teleworking and long-distance learning. While some services were adapted to tele-provision, many families did not have access to them at all [51]. While for those who did participate, the impact was dependent on a number of factors, such as age and degree of disorder and whether they were continuing therapy or just starting the therapeutic process [51]. A large proportion of these recorded minimal benefit, particularly for younger children, while caregivers of children with autism demonstrated the burden on the whole family [51].

In general, the delivery of support and intervention worked more effectively than the diagnostic process, which structurally involves face-to-face contact and necessary formal psychometric procedures [3]. Lindgren et al. (2020) [52] pointed out the potential of a successful distance training of parents in specific communication adaptability programs for young children with autism. For plenty of families and children with ASD, the exposure and use of technology are excessive, which implies their inability to manage the limits of appropriate internet use as well as the time of screen exposure, which for some families of children with autism was already a major problem [53].

The adverse educational circumstances under the coronavirus era were particularly unprecedented as special education schools, as well as mainstream schools, were closed down [54]. Most countries institutionalized school closure, and in some cases, tele-education was established [54] as another measure to limit social mobility. The closure lasted for months or even a year, especially for young children, Phelps and Sperry [55] point out that the focus of adjustments was on the restoring of the academic process online rather

than providing emotional and behavioral help and support for children in need. For some autistic children, online education while schools were kept closed was experienced with relief from stress and the fear of social criticism exposure through social interaction [46], but in other cases, many children experienced current change as a source of anxiety and stress. Both perspectives are reflected in research [47].

Although most parents found closure of schools and the parallel shift to tele-education as necessary measures, they nevertheless also highlighted their concern of children's academic failure and backsliding due to lack of motivation to participate in online education [56]. Although home education seems to be a rather innovative academic perspective in the Greek reality, it has gained research interest, especially as it concerns children with special educational needs (SEN) [26]. One such short qualitative research in five families of children with autism in the Philippines who were home educating children with ASD highlighted the need to involve all family members as there is a need for education in every form of activity in order to adapt to the new routine [57].

In the current study, we investigate how the quarantine and the subsequent government measures had a profound impact on the lives of young children with ASD and their families. The research questions were structured around two topics: containment measures and the provision of services. The key questions focused on how families and children with ASD were affected during the two extended periods of home confinement and the differences between the two. Additionally, how distance education and interventions functioned in the daily lives of children was investigated. The research focuses on collecting data and information in order to evaluate the deprivation of rights and accessibility of services for children with ASD as well as to shed light on the needs emerging in the new health crisis era.

The necessity and originality of the present study lie in the fact that it is an attempt to record adverse conditions as experienced by the families of children with autism in Greece. The unprecedented social condition of exclusion and confinement at home has overturned the international scientific dialogue in the direction of detecting the burden and the possibilities that open up in the context of the emergency [3]. The risks and impact of measures restricting personal freedoms and access to services have been highlighted and confirmed for the general population in numerous studies [58–60] but has also been particularly emphasized for at-risk groups, such as people with underlying or pre-existing conditions, like people with mental illness, elderly people, pregnant women, and homeless people [39,61–63].

Children and adults with ASD are a vulnerable social group in the current context, something that is not established internationally or by the Greek social and political formation, although the vulnerability of this social group to changes, stressful manifestations, and obsession with behaviors, events, and information has been considered as evident [64–67]. The current study, due to the lack of relevant studies in this context, is one of the first to be conducted in Greece—it attempted to contribute to the scientific knowledge by providing vital information collection and hopefully benefit families and children with autism [68] as well as educators, psychologists, logopathologists, occupational therapists, and social workers.

## 2. Methods

### 2.1. Study Design

The present research follows a qualitative research methodology, and the questions of the interviews were designed to gather data focus on "how?" and "why?" did residential confinement and universal lockdown affect the daily lives of families of children with autism in the context of the COVID-19 pandemic. The focus lies on the experiences of parents and children, and the researcher seeks the thoughts, feelings, and overall personal perception of the participants are investigated. Qualitative research designs and instruments, especially interviews, being flexible, offered the appropriate adaptability to the specific case and circumstance [69].

Data collection was conducted through individual semi-structured interviews [70]. The interviews were conducted by telephone in order to both comply with health measures and to fulfill the need to engage with the participants as much as possible under these circumstances [71]. An interview protocol was designed (see Appendix A) with the aim not to limit the personal narrative but to organize a productive discussion without the risk of slipping into a conversation outside the topic. The interview protocol is structured in two parts. In the first, demographic-type questions were included, which helped both to obtain essential information and to create an intimate atmosphere among the interviewees. In the second part, questions were asked to address the research question and purpose of the present study.

A pilot interview was conducted in order to obtain the appropriate feedback that would optimize the possibilities of the interview protocol and the researcher's ability to manage the interview process [71]. The researcher presented the interview protocol to a developmentalist to assess the relevance and extent to which the research objectives were met. Finally, to achieve the highest possible reliability during the interview, the questions were rephrased in a discrete manner to cross-reference the content of the conclusions drawn [72,73]. Based on the principle of openness and "active listening", sincere dialogue with the participants was promoted [73]. The interviews were recorded and transcribed based on a researcher-defined notational system, which was deemed necessary to faithfully convey the spoken word and oral expression in an attempt to avoid inaccurate subjective interpretation [74].

### 2.2. Participants

Participants were selected using the "snowball" or "chain sampling" method due to the difficulty of identification, as there is no overall registration of families with autism in Greece. Thus, some individuals were selected from the desired community and in turn provided access to other members of the population [75].

Below are the details of the participating parents (see Table 1). In total, 10 interviews were conducted with the mother caregiver (MC) or father caregiver (FC) of a child with ASD.

Participants were coded by interview number as N1, N2, . . . , N9, and N10. In listing the results, it was deemed necessary to register them with age and status in relation to the child; therefore, codes N1/43.PF, N2/42.PF, N3/45.PF, N4/45.PF, N5/49.PF, N6/52.PF, N7/45.PF, N8/33.PF, N9/49.PF, and N10/41.PF were formed. The interviews were 33 min long on average, with the interviews of the FCs being the shortest in duration (see Table 2).

Of the total of 10 interviews, three were conducted with FC (N6/52.FC, N8/33.FC, and N9/49.FC), while only one concerned a girl belonging to the spectrum (N7/45.MC). Most families had a change in the parents' employment status, with two of them losing their jobs (N4/45.MC) or being on suspension (N1/43.MC). Almost all families lost help and support from the wider family environment, particularly in P1, and relied solely on the abilities of both parents (N5/49.MC), while almost half of the families had daily support from special educators available online or over the phone (N3/45.MC, N4/45.MC, N9/49.MC, and N10/41.MC). Only two children were enrolled in a special school (N6/49.MC and N9/33.PF). Three of the families had only one child (the child with ASD) (N1/43.MC, N6/52.MC, and N10/41.MC), with one of them being a single parent (N10/41.MC).

**Table 1.** Demographic qualities of participants.

| | Family and Demographic Data | | | | | | | | | |
|---|---|---|---|---|---|---|---|---|---|---|
| Participants (interview number) | 1 | 2 | 3 | 4 | 5 | 6 | 7 | 8 | 9 | 10 |
| Age (years) | 43 | 42 | 45 | 45 | 49 | 52 | 45 | 33 | 49 | 41 |
| Caregiver relation | Mother | Mother | Mother | Mother | Mother | Father | Mother | Father | Father | Mother |
| Family status | Married | Married | Married | Married | Married | Married | Married | Married | Married | Divorced |
| Occupation | Private employee | Lawyer | Public official | Unemployed | Private employee | Private employee | Educator | Private employee | Private employee | Private employee |
| Manner of working in the period of confinement | In suspense | Rotationally | Rotationally | Not working | Remotely | Remotely | Remotely | Remotely | In person | Rotationally |
| Educational level | Higher education degree and higher | Higher education degree and higher | Higher education degree and higher | Private college degree | Private college degree | Higher education degree and higher | Higher education degree and higher | Higher education degree and higher | Higher education degree and higher | Higher education degree and higher |
| Number of children | 1 | 2 | 2 | 2 | 2 | 1 | 2 | 2 | 2 | 1 |
| Age of ASD child | 10 | 8 | 7.5 | 15 | 10 | 12 | 13 | 6.5 | 13 | 8 |
| Children's sex | Boy | Boy | Boy | Boy | Boy | Boy | Girl | Boy | Boy | Boy |
| Functioning level | High | High | High | High | Average-nonproductive speech | High | High | Average-nonverbal | High | High |
| School type | Typical | Typical | Typical | Typical | Special | Typical | Typical | Special | Typical | Typical |
| Educational parallel support | No | Yes | Yes (private) | Yes | No | No | No | No | Yes | Yes (private) |

**Table 2.** Interviews codes, duration, and date of conducting interview.

| Interview Code | Duration (Minutes) | Date |
| --- | --- | --- |
| N1/43.MC | 35:47:00 | 29 December 2020 |
| N2/42.MC | 43:41:00 | 2 February 2021 |
| N3/45.MC | 29:20:00 | 4 April 2021 |
| N4/45.MC | 36:59:00 | 11 March 2021 |
| N5/49.MC | 70:54:00 | 8 February 2021 |
| N6/52.FC | 19:55:00 | 11 March 2021 |
| N7/45. MC | 33:26:00 | 11 March 2021 |
| N8/33.FC | 26:04:00 | 23 March 2021 |
| N9/49.FC | 18:16:00 | 23 March 2021 |
| N10/41. MC | 38:52:00 | 1 April 2021 |
| Average time | 33 | |

### 2.3. Ethics

Ethical approval was granted by the Ethical Committee of Frederick University. This research is part of a dissertation thesis. All participants gave their informed written consent for inclusion in our study before they enrolled in the study.

### 2.4. Analysis

The interviews were analyzed using thematic analysis [75,76], which has been recognized as one of the best analysis methods in the context of social and anthropological research [77]. A distinctive characteristic of this method of analysis is that the data to be processed are not words but thematic-conceptual sets derived from them. The researcher is asked to identify meaningful sequences and repetitions that lead to the creation of codes [70,71,78]. The codes are synthesized and create autonomous conceptual units called Themes, which answer the research question; therefore, through the analysis, some main themes and sub-themes emerge [72]. We followed the six steps of thematic analysis proposed by Braun and Clarke [72]: familiarizing with the data, generating initial codes, searching for themes, reviewing themes, defining and naming themes, and producing the report. The illustration of the thematic analysis was chosen to be done through the creation of a thematic map [71]. In the data analysis, the relevant quotes are listed transcribed into written discourse [78].

## 3. Results

The thematic analysis resulted in three main themes and 12 sub-themes, which are related to the levels (themes) and mode/point of change (sub-themes) experienced by families of children with ASD. The main themes are: (a) the educational context (TH1), (b) managing the new reality (TH2), and (c) structuring everyday life (TH3). As sub-themes are identified: (a) backlash or adaptability (SUB-TH1), (b) school as a necessity (for the student and for the parent) (SUB-TH2), (c) new media possibilities and limitations (SUB-TH3), (d) the physical presence of a parent (SUB-TH4), (e) fear of illness and fear of the unknown (for oneself and for the other) (SUB-TH5), (f) the need for socialization (the personal need and the parent's anxiety) (SUB-TH6), (g) the teacher and the special educator (SUB-TH7)), (h) the educational process (SUB-TH8), (i) the parental needs (SUB-TH9), (j) women as employees and mothers (SUB -TH10), (k) the phases of home confinement (the differences) (SUB-TH11), and (l) new routine (activities, time, performances) (SUB -TH12).

### 3.1. Thematic Map

SUB-TH1 and SUB-TH2 occur in both TH1 and TH2 in a way that will be further discussed in each section. SUB-TH11 is a common component of Th1 and Th3. To explain this correlation, it is necessary to clarify that the distinction between P1 and P2 is not, in fact, a simplistic consideration based on the timing of the government's decision on universal closure. This exploratory process is not sufficient to understand the particular variations

and characteristics of each period, as there are individual differences and changes on an almost daily basis. At the same time, SUB-TH6 emerges as a context of both TH1 and a part of TH3. Finally, SUB-TH3 is a common thread across all themes as they are at the heart of the unprecedented condition of the pandemic. A table with a summary of the main themes and subthemes with a selection of quotes stated by the parents is included in Appendix B.

*3.2. Theme 1—The Educational Context*

The educational context refers to the educational process and the factors that consist of it, from the subjects to the place where it is carried out; these elements are modified in the context of the pandemic, forming a new condition. The sub-themes of this subject are identified as: (a) the backwardness and/or adaptability (SUB-TH1), (b) the school as a necessity (for the student and for the parent) (SUB-TH2), (c) the new media possibilities and limitations (SUB-TH3), (d) the need for socialization (the personal need and the parent's anxiety) (SUB-TH6), (e) the teacher and the special educator (SUB-TH7), (f) the educational process (SUB-TH8), and (g) the phases of home confinement (the differences) (SUB-TH11). It is argued that these codes constitute the formative educational framework that is crucial in the daily life of children with ASD.

3.2.1. Backlash and/or Adaptability (SUB-TH1)

By backwardness or adaptability, we mean the way and degree of response to the new academic and cognitive condition. On the whole, although students adapted to the new context, they encountered difficulties, and in some cases, there was a retreat of acquired abilities in their cognitive level and adaptation to school. In addition, many children showed stagnation and indifference in relation to school activity. The relevant quotes are presented below.

> "...setbacks as far as his responsibility is concerned, because now the lesson is in the living room and I am present because I do my activities, necessarily this is the place where we are both, so suddenly there was a little bit of logic "Mum, what did he say? What did the teacher say about tomorrow?" Similarly, I could see that he hadn't done some of the exercises, and I would point out to him the things that had struck me: "But your teacher told you that", so I was helping him and therefore that part of his own responsibility towards his obligations was lost..." (N2/42.MC)

> "With the beginning of the school year in the fifth grade, which is a demanding class, while we made a good start, we had a relative setback [...] they may not be in their place (school), and it is a big change but we do not have psychological fluctuations but he adapts very easily, I can say that the child adapted more than we did" (N1/43.MC)

> "Yes, yes it did them a great disservice, I don't know now how and what and whether they will continue the face-to-face courses or whether they will continue again behind a screen, I see that there is stagnation in {@}. Stagnation and indifference went into the phase of "I don't care" while until last year it was in the phase of "I should read and my ambition and not go without reading" (N4/45.MC)

3.2.2. School as a Necessity

As a sub-theme in Th1, the present captures the need or not as expressed by the children themselves to return to school (See Section 3.3.2). School was identified as a place of vital importance for children who longed for a return to school normality and found it difficult without it. On the other hand, and particularly for children with high potential and difficulty in social contacts, such as the majority of children with ASD, it was a trigger for further entrenchment in the personal boundaries of the self. In particular, some parents pointed out:

> "Returning to school was very pleasant, he was looking forward to it. Both were excited the day before. I think {@} must have slept for two or three hours out of anxiety to go to school" (N3/45.MC)

*"He liked the teleconference because he had his anonymity. So, imagine a child who is being severely bullied at school. This thing is taken to his home, in a very familiar place where he doesn't have the stress and anxiety, that he has to finish what he's been given in the 40 minutes of the hour and that he's given a deadline to finish it by the evening. And of course, he won't have the other person to troll him so to speak or I don't know what else he could do to him. He's in a safe environment and he tells you okay I like it and what's going on, it was something new, something cool perfect!"* (N4/45.MC)

*"He wants the schools to open since they opened and then they closed again. She wanted the schools to open so they could go, they were fed up of sitting at home and now I think it's a bit more intense"* (N6/52.FC)

### 3.2.3. New Media Possibilities and Limitations

In the new condition, contact with the educational reality was maintained by using new technology, both at the level of school and extracurricular activities and at the level of treatments and interventions, which utilized videoconferencing platforms and appropriately designed websites for the contact between specialists and the child. Most parents recognized the value of distance education as a solution of necessity that cannot replace face-to-face contact, while they pointed out their doubts about the effectiveness of distance education programs, treatments, and interventions. Some children, especially the older children with a high degree of adaptability, responded to the programs and made use of them. The same was not true for the children with greater support needs, and neither were the parents who attended the tele-education process. Additionally, difficulties with the use of computers were mentioned (see Section 3.3.3).

*"In the previous quarantine we had done a bit of intervention, speech and language therapy with the speech and language therapist online, but he did not participate he was watching himself and singing. He also did music therapy at a distance. Then, he would come, so basically, he only does the tele-education, he hasn't done any intervention remotely [...] This helped a little bit with contact, it doesn't replace school with anything, especially at these ages"* (N2/42.MF)

*"In the first quarantine, a program was piloted, I think, by the Stavros Niarchos Foundation we did a few lessons basically through some platform, his teacher and another person from the program side, more with games, so not purely educational. He doesn't do e-learning. He can't do it anyway"* (N5/49.MF)

*"With the child psychologist he does online even now and all the time, that is, for a year now he does online [...] it has not bothered him, maybe it is better because he doesn't have to go which is 20 minutes away by car"* (N6/52.PF)

*"At the beginning, we had some problems with the teacher until he learned the zoom, the codes, and all that. She kept losing the camera, losing the microphone, a lot of times we had problems. It's definitely a means for flexible forms of tele-education though, but it's definitely not immediate. {@} certainly doesn't help him, and I don't think it generally helps any child either"* (N3/45.MF)

### 3.2.4. The Need for Socialization

A key disadvantage of the absence of the school routine is linked to the possibility of socialization (see Section 3.4.2). For most parents, cutting off contact with the vital space of the school actually meant the loss of opportunities for socialization and meaningful contact with the "other". Other children expressed this need, and parents testified it in other cases distance, fostered the security of isolation. Parents reported:

*"It created some other issues, like not being physically present in their space with their friends and so on [...] so it was a different psychological state. Their friendships or activities had just been re-established and they lost them"* (N1/43.MC)

*"He definitely loses the contact with his classmates, he loses the playground in the yard, and he loses the contact with his teacher. So, for {@} what happens is not necessarily contact, because it's not really contact"* (N10/41.MC)

*"To a certain extent socializing with peers is definitely more relaxing when at home, that is, that part of WebEx she likes. I think it's not good for her, because it's something that pushes her on the one hand and on the other helps her develop. So, it would be good for her to be at school with other people. For her certainly the fact that she had that security of being in her own space where her environment is totally controlled by her is good, or she feels it's good"* (N7/45.MC)

### 3.2.5. The Teacher and the Special Educator

The persons who acted decisively in a positive or negative sense are the teachers. It is a common thread that most of them were unable to respond to the first unexpected P1. Nevertheless, they largely managed to gradually adapt, but no one raised the need to observe special measures for children with ASD, and there were cases where they were deprived of basic rights applicable in the face-to-face context. It is reflected that it is largely a matter of the individual teacher and his/her training and willingness to deal with the new media. To a large extent, parents who had SE at home note that it contributed significantly to a smoother adjustment to the period. Indicatively some parents report:

*"The teachers exceeded themselves, that is, even if they were not trained, I don't know if that was isolated or general. We were one of the cases where the woman had certainly not been in such a situation before, but she responded to the best of her ability, that is, the tools, I think, that were given were limited but they did a very good job"* (N1/43.MC)

*"Of the 15 or so teachers who could do the courses initially. 'Were there five? Who showed interest and joined the platform; the e-class? So many. Then slowly and now they are doing it more substantially [...] And it made a terrible impression on me, because basically he has the right in the exams if he wants to give orally and when he insisted—I was listening to him—that I want you to test me orally, he was told no through the WebEx that they were doing at that time"* (N4/45.MC)

### 3.2.6. The Educational Process

The educational process was completely different from the previous reality. In fact, in P1, there was no contact with formal forms of education and intervention for two months for a large number of children. Furthermore, it differed for children in typical education and children attending a special school, as shown in the SUB-TH11. The educational condition consists in forms of modern and asynchronous education, watching educational television, and extracurricular programs and activities. Students sometimes with the presence of a special educator or parental intervention and sometimes without attended language classes in addition to school. As reported by parents:

*"He had the security of not being seen, because they didn't have cameras on, let me start from that, he would get up, he would twist around, he would do things that the security of not being seen provides [...] it was an intermittent lesson with a lot of breaks and the teacher not having the rhythm required. As a result, time was wasted like that. [...] Most kids are free to do whatever they want because they are not seen"* (N1/43.MC)

*"In the school at the beginning when the project started one was working webex one was not. The teacher didn't make them do anything for homework, so whatever they did (in class) with his teacher and in the private tutoring lesson"* (N6/52.FC)

*"Because they were given the opportunity to do any combination they wanted or webex and eclass or webex only or eclass only they had to attend all at the same time, and it was a bit difficult to organize themselves a bit. It was definitely easier for her in the webex courses because once you had to follow something she was given clear instructions on*

*what to do next, whereas in eclass she was a bit struggling with what she had to do and how to do it and she needed a bit of help from us*" (N7/45.MC)

### 3.2.7. The Phases of Home Confinement

The different phases of the pandemic and the modified measures with the gradual progression can be distinguished in two essential phases for the school reality in period P1 children were found without any form of education and support with the only possibility of distance intervention SUB-TH3. This period was a trial period for teachers and pupils and was followed by P2, during which formal schools operated exclusively remotely while special schools operated face-to-face. P1 was a critical period for children needing more support, while for others, it was simply a break from a busy daily routine. In P2, the special services are now accessible, and this creates a more favorable condition.

"*In the first period, there were no teachers, all the teachers were suspended. They had set up a Viber from the nursery where each teacher sometimes sent an activity. So, in the first period, that is until June when the schools opened for a while, there was no school activity*" (N10/41.MC)

"*At the beginning they saw it as a holiday and a break in the routine of the school, because the first quarantine found us at the end of the season, the school term, while the second one was at the beginning and they were not projected to be so tired or bored or whatever*" (N1/43.MC)

"*The first quarantine was much more difficult because the special center we go to and work with occupational therapist and speech therapist was closed, meaning no sessions at all. Like the other center that he goes on weekends and after school. So, he had no activities he stopped everything. Towards May, he started the center that does the after-school programs. [...] so there he slowly got into a little bit of a rhythm and started to have contact like that. Whereas now we do regular and speech therapy and we move with one and we take him [...] so it is much better in the child's matter this time, because the child experiences it as if there is nothing different. It is like there is no change in the program*" (N5/49.MC)

### 3.3. Theme 2 Managing the New Reality

Managing the new reality includes the reactions, emotions, thoughts, and overall elements that determine the attempt to adapt to the new reality on the part of parents and children. The sub-themes of this subject are identified as (a) the backwardness and/or adaptability (SUB-TH1), (b) the school as a necessity (for the student and for the parent) (SUB-TH2), (c) the new media possibilities and limitations (SUB-TH3), (d) the physical presence of a parent (SUB-TH4), (e) the fear of illness and fear of the unknown (for oneself and for the other) (SUB-TH5), (f) the parental needs (SUB-TH9), and (g) women as employees and mothers (SUB-TH10).

### 3.3.1. Backlash and/or Adaptability

The present is also identified as a sub-theme in TH1 (see Section 3.2.1), as there was a repetition of statements in relation to children's broader improvement or regression in various areas of daily human life. For the most part, the children adjusted. They did, however, encounter a high degree of difficulty either in the first particularly unexpected and unfamiliar P1, and the situation had a more negative impact with more time spent in isolation at the end P2. Although no emergencies were noted, parents indicated the discomfort and resentment expressed in a variety of ways and also progressions that occurred due to the extended stay at home. They indicate:

"*The child had explosions in the first quarantine and we had an explosion almost every Monday. Of course, gradually they became less and less intense, at some point the child even exploded and expressed it verbally, which they thought was very good. He said that I want to go to school, I want to go to see my friends, I want to go to {@} to go to the*

*shops I like [...] while he had some difficulties, he ate very little food and now he eats much more*" (N2/42.MC)

"*His daily routine has changed, his routine has changed and his behavior has changed, he is more hyperactive in the house with a lot of voices, communicative and non-communicative, because there are voices that communicate with you loudly and there are voices that are a tension, a release of energy [...]. Generally, in phases of exacerbation, he was in a tension that would not calm down with anything [...]*" (N8/33.FC)

"*He was going on YouTube, and he had to see what was happening in relation to other countries, what phase they were in, what their mortality rates were, how many cases a day, he has also entered this phase, that is, in a kind of obsession of what is happening over there. Okay, we have overcome this, it may come and go, we have our bad and good days*" (N4/45.MC)

### 3.3.2. School as a Necessity

School was not only a necessity for the students (see Section 3.2.2) but also for the parents themselves, in their efforts to manage their own and their children's personal, work, domestic and individual needs. School, as a structure and as a process, gives parents the necessary space and time to work and meet wider needs, but it is also a safety net and developmental network for the child. They point out:

"*It was very difficult, I mean in the second quarantine until I heard that the special schools would remain open, I was really, really sick, until I heard the announcement that is, I couldn't imagine how the child could go through all that again*" (N5/49.MC)

"*I take as a positive that they made sure that at least for the second period the special schools stayed open. I think that it was at least, from the point of view of the state, something, a sign that they are thinking about some families, they are thinking about some people*" (N8/33.FC)

### 3.3.3. New Media (Use of Technology) Possibilities and Limitations

A key point highlighted by parents was the need to delimit the use of new media, which became difficult since the school curriculum and every other educational activity included the use of a computer, tablet, or smart device with internet access (see Section 3.2.3). Parents noted:

"*In general, we are not very much in favor, not of technology, of use by children I mean. So, I try to have a limit to that which of course is violated as a principle because of tele-education [...] over exposure to a screen, because it's not only school hours it's also English that he does and that becomes a means of tele-education. So, there are days when he can do five and six hours in front of a screen*" (N1/43.MC)

"*The little boy wants to play with his games all the time, which we don't want him to play so much in the week, that is with PlayStation and with tablets and that's where we have some fights. And he, from his side, says and what should I do? Since he can't see his friends, he doesn't have sports, he doesn't have those and there is so little in it, it's a heavy atmosphere*" (N6/52.FC)

### 3.3.4. The Physical Presence of the Parent

The physical presence of the parent in home life, and particularly of the MC, was a key factor in managing the new reality, both for the children and for the parents themselves. Parental presence replaced help from the wider family environment. The same condition also has the negative aspect of redundancy of shared space and time from a certain point onwards, as times are compressed, which is not always discernible to the child. They mention:

"*The positive aspect of the whole thing is that I had the time. In other circumstances, normal circumstances, I would have been away from the child. I had the time to be*

*very close to him, as a physical presence, which was the only positive thing in the whole management of the period [...] as physical presences my husband and I were almost constantly together and with the child. In other circumstances when I would have been working, my husband and I would have definitely had support from our parents, i.e., grandmothers, grandfathers to hold the little one now we didn't even need"* (N1/43.MC)

*"We spent a lot of time together as a family, so we had good times [...] I find it hard to set boundaries with my children but because I spent so many hours with them, I had the peace and patience. It was different, it's different to be at home"* (N2/42.MC)

### 3.3.5. Fear of Illness and Fear of the Unknown

A key concern expressed by parents and children related to the fear of getting sick themselves or their loved ones. A pattern of anxiety in relation to the unexpected or the unknown was also identified. This element of fear determined the attitude of families during the first period.

*"Because out of fear and we were also restricted. We did not do anything"* (N8/33.FC)

*"He expresses fears that I don't want anything to happen to you, I don't want you to get infected and all that, but I don't think he has it in his mind all the time. I think it's more an expression of I love you, I don't want anything to happen to you, rather than it's so deeply ingrained in him that it guides his behavior, he's careful of course"* (N1/43.MC)

*"He was saying his tummy hurts and he wants to vomit and he said: "I'll wear the mask, I'll wash my hands, I want to be safe, I don't want to catch viruses, I don't want to catch coronavirus"* (N2/42.MC)

*"You have the anxiety: what is the child going to do; how is he going to cope? That anxiety let's say from the first quarantine and it stays with me and I think with my husband, there is this insecurity that how will the child cope if I get sick and have to be away for so many days?"* (N5/49.MC)

### 3.3.6. The Parental Needs

A plethora of needs emerged through the parents' narratives, with the dominant one being the desire to access structures and the sense of having a supportive environment and help. On occasions, the need for personal space and time to absorb the stresses was identified.

*"We understood that we can manage anyway. I mean without saying, we might not have gone to the center but it was supportive yet and that helped us because I realized that I have some people who won't leave us in difficulty"* (N10/41.MC)

*"Now I feel it's a bit too much, this thing, I mean I want to say to them, go away everyone go to your school, go to your work, leave me a bit to have some time and I ((laughter)). Be left alone. Not listening to anybody. [...] To be honest, whatever we say you want your privacy too. You want to say something with someone else, you want some privacy. That's how it is, the whole thing is needed"* (N4/45.MC)

### 3.3.7. Working Mothers

Another feature of the period concerns the burden placed on MCs. The levels affected are three. In the beginning, it was recorded that in most cases, domestic work is entirely their own business. In a period when the work and home space is unified, this is intertwined in a special way. At the same time, some of them experienced the loss of their working identity as a result of the existing condition with a psychological and emotional burden. As it has been pointed out so far in every issue, there are positive and negative perceptions.

*"As a working woman I was completely out of it, but it was simply compensated by the fact that for the first time in history I was with my son all the time"* (N1/43.MC)

*"When you're in the office, you're in the office; when you're at home, you're at home. I mean, it's a bit more shared, when you have to do them somewhere at the same time,*

*I mean in parallel not at the same time it's much more difficult. That made me tired, honestly it made me tired, it broke me down I can say"* (N3/42.MC)

*"It's not easy in the new role I've taken on, because in October, I took over as managing director in the company, it wasn't easy to maintain telecommuting. [...] So it was a period when I had to combine {@} at the same time. {@} to have a full day but also for me to be able to follow exactly what was going on at work. [...] So basically, I was trying to spend the hours when I didn't have {@} to get on with the work the hours when I had {@} to have a schedule with {@}. What I couldn't control were the phone calls. It happened during various activities, various phone calls would pop up from work, some were short some other not that short, so there I was trying a little bit to change the activity we were doing with one that didn't require so much of my involvement"* (N11/41.MC)

### 3.4. Theme 3 The Structuring of Everyday Life

The structuring of everyday life in the context of enforced house arrest was a universal concern for all families who were called upon to formulate a new daily routine and routine adapted to the new measures and health restrictions. For most families, this was a great difficulty, especially in P1, which was an unfamiliar situation, and there was neither preparation nor assistance.

The sub-themes of this subject are identified as: (a) new media possibilities and limitations (SUB-TH3), (b) the need for socialization (personal need and parent's anxiety) (SUB-TH6), (c) the phases of home confinement (the differences) (SUB-TH11), and (d) new routine (activities, time, performances) (SUB-TH12).

#### 3.4.1. The New Media: The Possibilities and Limitations

As noted in previous relevant sections (see Section 3.2.2), technological resources and the internet were a pillar of life-building in the context of the pandemic. Typically, it was an aspect that offered potential but also upset previous balances on children's use of it (see Section 3.3.3). Parents point out:

*"There are too many hours on the computer, you constantly have the feeling that you are working and you have no outlet afterwards. And she's complaining this year, which she didn't complain last year, that I'm always studying, that I don't have time to do the things I like—well, she says that all the time, of course—but laughing, I don't have time to do the things I like, I don't have time to play, I don't have time to watch YouTube, I don't have time to search the internet for things that interest me, I always have to read"* (N7/45.MC)

*"Thank god we have internet ((laughter)), because if we didn't have internet, we would have . . . imagine three days we didn't have internet and it was really difficult. With {@}, with classes and with work it was really hard. The necessity of the internet became apparent"* (N6/52.FC)

#### 3.4.2. The Need for Socialization

Socialization was a constant concern of parents and an expressed desire of children. In some children, the need to socialize was born while others were further confined to the safety of home. For many children, this need was also reflected in their relationship with their siblings (also see Section 3.2.4).

*"In the second phase after Christmas and from a certain point onwards in both the first and second quarantine, we did as much as possible excursions for walking so that they could meet friends. [...] he saw that there is this way and he could no longer close himself off. And so now with this knowledge the second quarantine is over, and he wants to go out whenever the weather allows and meet children"* (N1/43.MC)

*"So he has children that he is more connected to than others and he mentions to me and he also wants us to invite them home. He has asked me [...] I want to hug my {@} (Brother's name) my {@} my {@} my little boy I want to go downstairs I want to go with {@} [...]*

*but I didn't see any difference in their relationship, on the contrary I can say that they are developing better because I think that both of them are growing up so the play and interaction acquires another quality"* (N2/42.MC)

*"In relation to her friend, I won't say her friends because it was one. They didn't communicate so much; they didn't see each other almost at all. Now, in the second quarantine, I see that it is different; there is communication with Viber, there are calls, there are always text messages, we arranged and met and once we had gone for a walk. Now, we are trying to arrange something like that again"* (N7/45.MC)

### 3.4.3. The Phases of Confinement

For the majority in P1 for high functioning children, it was a form of interruption; it was not the same for children in need of more support. The emotional state and attitudes of the subjects varied according to the level of routine mastered and the child's level of functioning. Although the comparison of P1 and P2 is not clearly nuanced, it is clear that the prolongation of the P2 period signaled parent and child exhaustion.

Parents point out:

*"In the first quarantine I think the work was not particularly demanding for me or my husband so it didn't put a strain on our schedule. But now in the second quarantine, everything was working, everything was working normally unfortunately and the fact that we were half there then we found twice as much work ahead [...] the second quarantine was definitely rather exhausting in terms of work"* (N4/45. MC)

*"The first with the second quarantine has differences, and the child himself expresses them. I think in general and personally that we were burned in the first quarantine, which was the first shock, so now the second quarantine is more difficult, at least that is how we experience it"* (N1/43.MC)

*"The first quarantine was, I guess you have heard it, and from other parents it was a nightmare [...] We started to take him out after they found out after a while ((ironic tone)) that there are also children with special needs where we had relatively more freedom of movement [...] that is, it is much better for the child this time, because the child experiences it as if there is nothing and there is no change in the program"* (N4/49.MC)

*"This one is much more difficult psychologically. In fact maybe because now there is a very strict schedule, which leaves no room for relaxation, because in the previous quarantine, to be honest, we all had a little less work [...] The truth is that it is a much more constricted schedule given that it is 8:30–14:00 most of the time plus her English, plus her support, which she goes normally, because she hasn't stopped, so it is a very structured schedule now"* (N7/45.FC)

### 3.4.4. The New Routines and Performances

The new routine is defined as the new daily routine and the way of organizing in the light of home restraint measures. To a large extent, this routine was permeated by the health measures, of which wearing masks and keeping distances were the measures that families were largely unable to implement. Despite the attempt to establish a new routine, the desire to return to the previous condition continues to be a demand of children and parents.

*"He had expressed many times the question when will I get my life back; that was exactly the phrase I'm not telling you metaphorically "mum, when will my life come back"* (N3/45.MC)

*"In the second quarantine from a certain point onwards the mask became compulsory everywhere. There was a grumbling when the mask was only in the classroom, but she could take it off at break time well it was even more grumbling once it was done: from the time I left my house until the time I got home. And it's still a problem, meaning many times when we try to get her to leave the house her nagging is: "Yeah but I don't*

*like wearing the mask and you know it. Okay we'll go out to walk, but first of all where are we going? and secondly, I don't like wearing the mask and walking. I don't like the mask!"* (N7/45.MF)

*"He lacks physical exercise. He misses it a lot and it makes him tense and he can't sit quietly afterwards in the house. That is, he goes crazy. We have got him a basketball here; he plays an informal basketball here"* (N3/45.MF)

*"It's very specific routine, it's the school routine. You will get up in the morning you will start your lessons at a certain time, you will finish them at a certain time, you will take a break to rest a little bit; you will start studying, which will go until you finish studying and then you can do whatever you want. And in the evening, there is usually a time when we all sit together, we eat together, we watch TV, we talk, a family, more family part"* (N7/45.MF)

## 4. Discussion

The thematic analysis of the data revealed, as expected, a profound change in the daily life of children with autism and their families.

One of the main findings of our research was that despite the fact that the children showed a high degree of adaptation and adaptability—either they perceived the condition or simply experienced it—they showed a series of emotional, psychological, and behavioral changes, which resulted primarily from the suspension of the familiar routine [66] This component is core to the characteristics of the spectrum and was the main source of stress and anxiety for parents, who had to take over entirely the formation of the new daily routine [45,79]. In addition to this, they were also required to substitute the role of specialists, particularly in P1, where access to services and school was suspended for a total of two months, a fact that has been mentioned by other researchers [35,36,56].

Additionally, our participants highlighted the importance of experts in their lives—teachers, special educators, and psychologists who contributed to the normalization of the condition [80]—as noted for families with shared experiences or friends who kept in telephone contact with caregivers. For parents of our sample, support, even at the simple level of concern, was crucial for their positive development and psychological empowerment, a finding that is similar to previous studies [57,58,81,82].

According to parents of our sample, their children experienced psychological and emotional changes manifested by outbursts, intense hyperactivity, the formation of new stereotypes and obsessions related to home confinement and the progression of the pandemic; see also relevant results from other studies [61,65]. Problems with sleep and food preferences were also noted [60,83,84]. Berard et al. [85] assessed that single-parent families show a greater breadth of impact of measures with positive and negative implications, which is related to the parent's workload and which was confirmed in the present study. In our research, most children had less difficulty in staying at home, at least in the early stages of the pandemic and more with the measure of mask use and social distance.

As for the comparison of the two periods of confinement, the parents of our study differ both in terms of the functional factor of children with ASD and in terms of management and parental attitudes. Parents in P1 experienced intense fear and anxiety about the new period, which could be assuaged by the reduced workload, limited measures, and the light holiday atmosphere experienced by high-functioning children [54]. P1 was a critical period for children needing more support (N5/49.MC, N.8/33.FC) as the complete lack of support created some setbacks and predominantly anxiety and stress for parents about how to manage the new reality [43]. The existence of open structures and special schools in the second period in Greece was crucial for all children and especially for parents as it freed up some of the time they could spend on work or personal time. A condition that is a persistent problem for parents of children with ASD [86].

Our research showed that the distance learning process worked for the high-functioning children, but it gradually led during P2 to the birth of a tendency of indifference and disinterest [56], as well as to the feeling of a generally stressful daily life without any outlet

for relaxation and rest, a finding that is consistent with the results of research in the neurotypical population [60]. At the same time, there was a significant decline from parents in the rights of children integrated in the mainstream classroom but also an absence of individualized instruction and meaningful support from parallel support teachers as this was not possible in the online classroom.

The closure of face-to-face services opened up the great chapter of tele-medicine and tele-interventions, procedures that all families of our study have tried. For most, as research confirms, they are ineffective, but their use recommends a necessary countermeasure to the complete absence of centers [51,59]. In terms of socialization for a large proportion of children of our sample, it was completely lost in the light of not interacting at the school level or other activities. The most typical finding, which is fully supported by both the literature and contemporary research, relates to older children who became more self-contained and accepted the security of home as the appropriate environment where emotional, psychological, and social shocks are not experienced [46,87]. At the same time, for most children of our sample, online communication was only a solution if there was a previous background of communication. What became clear is that children with ASD want and crave communication and friendly interaction but do not master how to perform this social act [88].

However, there were also signs of improvement or progression due to increased time spent interacting with parents in terms of specific skills, especially in the area of communication and autonomous activity management, according to parents [85]. In fact, the improvements were mainly noted in younger children, a finding confirmed by contemporary research and literature [85,89]. Working parents may have experienced more intense stress, but they also regained time to interact with the child, which gave them both confidence and the ability to better manage. Finally, a key concern for parents of our study is how to impose limits on technology use in a context where long hours of screen time imply fulfillment and participation in the educational process [53].

### 4.1. Limitations of Our Study

The limitations of our study are that our participants were not randomly selected, and we based our research on a small group of parents. Secondly, telephone interviews may be an "easy" way to gather data, but they have several disadvantages due to the lack of the face-to-face interaction. Finally, we could enhance our questions and include questions that could provide us with more "in deep" answers.

### 4.2. Suggestions for Future Research

From the scope of the proposals and the limitations of the research, aspects and dimensions that need to be studied in depth are suggested. Firstly, it is necessary to collect data at the national level. At the same time, research on the level of web-based tools and applications is proposed in order to identify ways to use them more effectively. It also seems important to look for further implications and dimensions of the existing condition for both low-functioning children and adults with autism. This proposal taps into the need to outline overall the importance of confinement and emergency for people with autism in order to improve accessibility, services, and, overall, their daily lives.

### 5. Conclusions

The results of the study confirmed the initial hypothesis of a change in the daily routines of families and children with ASD; with the disruption in daily established routines and interventions being the main factor that caused and increased stress in the children's caregivers. At the same time, a significant impact of the measures on the already disrupted work-life balance of caregivers was illustrated. Parents were already experiencing a condition of social isolation due to the increased needs of the children and social stigma, but the condensation of time and activities in the single space of the home without outlets of release produced a further aggravating condition, an element that

highlights the necessity of psychological and material support for parents [44]. Government should develop an organized plan for an emergency situation (such as the pandemic) to support parents and their children with ASD in order to cope with the obstacles that they may arise and the high levels of stress which is emerging. Professionals should use telehealth platforms in order to evaluate the individualized needs of every child and develop the appropriate actions for crisis management. Teachers should pay attention to including parents as co- therapists to tackle the emerging situation and the immense stress their children and themselves are facing from the pandemic.

It is clear that the present research could not and did not seek to cover the full range of limitations and possibilities that open up in this new period but to highlight the level of change in everyday life of families and children with ASD, as a first collection of material to identify overall directions for research. Thus, three essential aspects of defining the new everyday life were reflected, with the new means of intervention as the central pillar, followed by the change in the level of the way parents manage the new reality and finally the overall change in the educational process and interventions. The results of the research confirm the broadness and heterogeneity of the spectrum, the crucial role of the parent caregiver [90] in the child's manifestations and reactions, and the necessity of support programs and school in the daily life of children with ASD. The impact of the pandemic cannot be judged to be exclusively positive or negative. The common thread of all was that the long-term imposition of the forced home confinement regime exacerbates psychological stress for parents and students [35]. All this confirms that the end of the crisis will lead to the beginning of an extended phase of post-traumatic shock [37], but there are also possibilities, especially at the technological level, which ultimately mean necessary advances in adapting to the new period.

**Author Contributions:** Conceptualization, M.S. and S.P., methodology, M.S., M.G., software, M.S..; validation, M.G.., S.P., D.M.T. and N.L.-I.; formal analysis, M.S., M.G., S.P..; investigation, M.S.; resources, M.S., M.G..; data curation, M.S., S.P., M.G., D.M.T., N.L.-I.; writing—original draft preparation, M.S.; writing—review and editing, M.G., S.P., N.L.-I., D.M.T.; visualization, M.S.; supervision, S.P. All authors have read and agreed to the published version of the manuscript.

**Funding:** This research received no external funding.

**Institutional Review Board Statement:** Ethical approval was granted by the Ethical Committee of Frederick University. Ethics Committee School of Education and Social Sciences, University of Frederick Approval Code: BE_Saliverou_15/10/2020.

**Informed Consent Statement:** Informed consent was obtained from all subjects involved in the study.

**Data Availability Statement:** The data presented in this study are available on request from the corresponding author.

**Acknowledgments:** The authors would like to thank all the participants of our study for the willingness to participate in our research. All individuals have consented to the acknowledgement.

**Conflicts of Interest:** The authors declare no conflict of interest.

**Appendix A**

*Interview Protocol*

1. *COVID-19 information*: How did you inform (your child) about COVID-19?

    a. Did you have any help (school, therapist)?
    b. What procedure did you follow?
    c. Did he understand it literally metaphorically?

2. *The new everyday life*

    a. Is it difficult to apply the measures in your daily life with him/her . . . does he/she
    b. understand them, e.g., about wearing a mask? Did it make it difficult for you?

c.   Have you found ways to structure the daily routine at home?

3.   *Interventions—counseling*: During quarantine, did you have access to counseling/interventions and health care services?

   a.   Did you follow a regular program, or were you recommended a structured program?
   b.   Did you experience any emergency situations?
   c.   How do you evaluate teleconferencing and providing such services online?

4.   *School:*

   a.   In relation to his/her school activities, did they continue as normal (synchronous or asynchronous tele-education)?
   b.   Is he/she attending his/her classes and doing his/her homework daily as normal?
   c.   How has this worked for him/her? Did it help him/her in his/her daily routine?
   d.   Did he/she express emotional tensions (anxiety, stress, nervousness, tantrums) when the child had to teleconference?
   e.   Do you feel that homeschooling is now an alternative?
   f.   The fact that special education schools are open is comforting? (Has it helped you that special education facilities are open?)
   g.   Do you feel the teachers were adequately trained and able to help? (although special schools were open during this quarantine, not in March, but some were closed due to outbreaks)

5.   *The child and the quarantine:*

   a.   What was your daily routine?
   b.   Did other family members take on roles?
   c.   Would you say that since the previous quarantine, the situation has been normalized?

6.   *The announcement of the second wave and restraint measures?*

   Are things better now?

   a.   With the school 'or with special therapists (psychologist, speech therapist), did
   b.   you choose to attend teleconferences or face-to-face services (education/therapy)?
   c.   Any difficulties you had experienced in the previous quarantine were
   d.   smoothed out, and what contributed to adjusting to this one?

7.   *Access to public or state services, benefits*

   a.   The digitization of the provision of services of the CEDAs

8.   *The experience of getting sick*

   Did you or someone close to you get sick, and how did you experience it?

9.   *General emotional state*

   Which were the most difficult feelings that you experienced?

10.  *Internal or external support*

   What helped you meet the increased and unprecedented demands of everyday life?

11.  *The general experience*

   How are you experiencing the situation, and how does this whole quarantine period affected you?

12.  *General emotional effects on their lives*

   Ultimately, do you feel that you were positively or negatively affected by the extended stay at home? In which way?

## Appendix B

**Table A1.** A selection of quotes for each theme.

| | Educational Framework | Managing the New Reality | Structuring of Everyday Life |
|---|---|---|---|
| Backlash or adaptability | *"With the beginning of the school year in the fifth grade, which is a demanding class, while we made a good start, we had a relative setback [...] they may not be in their place (school), and it is a big change, but we do not have psychological fluctuations, but he adapts very easily, I can say that the child adapted more than we did"* (N1/43.MC) | *"The child had explosions in the first quarantine, and we had an explosion almost every Monday. Of course, gradually, they became less and less intense; at some point, the child even exploded and expressed it verbally, which they thought was very good. He said that I want to go to school, I want to go to see my friends, I want to go to {@} to go to the shops I like [...] while he had some difficulties, he ate very little food, and now he eats much more"* (N2/42.MC) | |
| School as a necessity | *"Returning to school was very pleasant; he was looking forward to it. Both were excited the day before. I think {@} must have slept for two or three hours out of anxiety to go to school"* (N3/45.MC) | *"It was very difficult, I mean in the second quarantine until I heard that the special schools would remain open; I was really, really sick, until I heard the announcement that is, I couldn't imagine how the child could go through all that again"* (N5/49.MC) | |
| New media possibilities and limitations | *"In the previous quarantine, we had done a bit of intervention, speech and language therapy with the speech and language therapist online, but he did not participate; he was watching himself and singing. He also did music therapy at a distance. Then he would come, so basically, he only does the tele-education, he hasn't done any intervention remotely [...] This helped a little bit with contact, it doesn't replace school with anything, especially at these ages"* (N2/42.MF) | *"The little boy wants to play with his games all the time, which we don't want him to play so much in the week, that is with PlayStation and with tablets, and that's where we have some fights. And he, from his side, says, and what should I do? Since he can't see his friends, he doesn't have sports, he doesn't have those, and there is so little in it, it's a heavy atmosphere"* (N6/52.FC) | *"Thank god we have internet [laughter], because if we didn't have; internet we would have . . . imagine three days we didn't have internet and it was really difficult. With {@}, with classes and with work it was really hard. The necessity of the internet became apparent"* (N6/52.FC) |

**Table A1.** *Cont.*

| | Educational Framework | Managing the New Reality | Structuring of Everyday Life |
|---|---|---|---|
| The need for social-ization | *"He definitely loses the contact with his classmates, he loses the playground in the yard, and he loses the contact with his teacher. So, for {@} what happens is not necessarily contact, because it's not really contact"* (N10/41.MC) | | *"In the second phase after Christmas and from a certain point onwards in both the first and second quarantine we did as much as possible excursions for walking so that they could meet friends. [...] he saw that there is this way and he could no longer close himself off. And so now with this knowledge the second quarantine is over, and he wants to go out whenever the weather allows and meet children"* (N1/43.MC) |
| The phases of home confine-ment | *"The first quarantine was much more difficult because the special center we go to and do work and speech was closed, meaning no sessions at all [ ... ] so it is much better in the child's matter this time because the child experiences it as if there is nothing different. Like there is no change in the program"* (N5/49.MC) | | *"This one is much more difficult psychologically. In fact, maybe because now there is a very strict schedule, which leaves no room for relaxation, because in the previous quarantine, to be honest, we all had a little less work"* (N7/45.FC) |
| Educational process | *"He had the security of not being seen because they didn't have cameras on, let me start from that, he would get up, he would twist around, he would do things that the security of not being seen provides [...] it was an intermittent lesson with a lot of breaks and the teacher not having the rhythm required. As a result, time was wasted like that. [...] Most kids are free to do whatever they want because they are not seen"* (N1/43.MC) | | |

**Table A1.** *Cont.*

| | Educational Framework | Managing the New Reality | Structuring of Everyday Life |
|---|---|---|---|
| Teacher and special educator | *"The teachers exceeded themselves, that is, even if they were not trained, I don't know if that was isolated or general. We were one of the cases where the woman had certainly not been in such a situation before, but she responded to the best of her ability, that is, the tools, I think, that were given were limited, but they did a very good job"* (N1/43.MC) | | |
| Physical presence of parent | | *"The positive aspect of the whole thing is that I had the time. In other circumstances, normal circumstances, I would have been away from the child. I had the time to be very close to him, as a physical presence, which was the only positive thing in the whole management of the period [...] as physical presences my husband and I were almost constantly together and with the child. In other circumstances when I would have been working, my husband and I would have definitely had support from our parents, i.e., grandmothers, grandfathers to hold the little one now we didn't even need"* (N1/43.MC) | |
| Parental needs | | *"Now I feel it's a bit too much, this thing, I mean I want to say to them, go away everyone go to your school, go to your work, leave me a bit to have some time and I ((laughter)). Be left alone. Not listening to anybody. [...] To be honest, whatever we say, you want your privacy too. You want to say something with someone else; you want some privacy. That's how it is; the whole thing is needed"* (N4/45.MC) | |

**Table A1.** *Cont.*

| | Educational Framework | Managing the New Reality | Structuring of Everyday Life |
|---|---|---|---|
| The fear of illness and/or for the unknown | | *"You have the anxiety: what is the child going to do; how is he going to cope? That anxiety, let's say from the first quarantine, and it stays with me, and I think with my husband, there is this in-security that how will the child cope if I get sick and have to be away for so many days?"* (N5/49.MC) | |
| Working women as mothers | | | *"When you're in the office, you're in the office; when you're at home, you're at home. I mean, it's a bit more shared, when you have to do them somewhere at the same time, I mean in parallel not at the same time, it's much more difficult. That made me tired, honestly, it made me tired, it broke me down I can say"* (N3/42.MC) |
| New routines and performances | | | *"Now what he can't no matter what is the mask because any-way with his head he has some tactile issues, so in his head he doesn't want much of any-thing. On Halloween, when he dresses up, he doesn't want hats or the summer hat, which we wear this with a lot of ef-fort now"* (N10/41.MF) |

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
