# Peer review of "The Impact of COVID-19 Pandemic Containment Measures on Families and Children with Moderate and High-Functioning ASD (Autism Spectrum Disorder)"

_education, doi:10.3390/educsci11120783_

Round 1

Reviewer 1 Report

Dear authors,

thank you for an interesting read. The article starts with a good literature review specific to the impact of the pandemic on children with autism and their families. Some more detailed discussions would be recommended (see below). Methodology is clear. Results provide very important direct quotes from the parents explaining their specific situation during the lockdowns. Discussion summarises the results and links them to previous research studies. Overall a very interesting article. With some small changes and a check of English it will be an excellent contribution to current knowledge in the area.

Notes:

Line 30-31 Why does autism present a challenge in the pandemic? Would not it be better to phrase it that the pandemic presents even a more difficult challenge for people with autism and explain why.

Page 2 talking about the pandemic – I would recommend using past tense

Line 68 “The findings of research on the difficulty of achieving balance in the family of a child with autism [21], combined with the broader picture of special education in Greece [22,23],” Could you be more specific on both points?

Line 121 Grammar: rephrase: “whether they were new to treatment or people who were continuing a therapeutic intervention [47].”

Line 126 “diagnosis” change to “diagnostic process” or “assessment”

Line 134 “typical” schools change to “mainstream”

Line 133-135 It would be useful to specify how many children with ASD (or what percentage) attend special schools.

Line 160-161 “accessibility of children with ASD” perhaps “accessibility of services for children with ASD”

Line 171 “mentally ill,” rephrase “people with mental illness”

Line 179 only educators? perhaps also other service providers

Line 183 use past tense

Line 228 Table 2 Missing number in dates

Line 236 special educator located at home – unclear whose home? maybe special educator available online or over the phone?

Line 237 “Only three families were families with one child with ASD” It is not clear if it means they had other children and only one of them had ASD, or if they only had one child. Rephrase: The child with ASD was the only child. Overall, in the participant table it is not clear if the families only had one autistic child (it looks like it), but families often have more than one autistic child.

Table 3 Does not provide any new information. It is not necessary.

Line 278 Thematic map – I find it unnecessary. This is not a thesis. It is enough to state what the themes and subthemes were as you did in the paragraph before.

Line 296 Table with quotes. Again unnecessary. The quotes are presented later in the results and that is enough for this study. Also, some quotes could use checking for translation (e.g. explosion – meltdown).

Line 311 Thematic map is unnecessary. It just repeats the information that was already said in the text.

Line 489 Here the subthemes are not listed in the text, so the figure makes sense, however, I would recommend the dame approach in all themes discussed, preferably, listing them in the text. (Same for Theme 3).

Line 502 What do you mean by deepening and expansion of the condition? It seems to be suggesting that their autism is getting worse. Maybe rephrase that the situation had a more negative impact with more time spent in isolation?

Line 599 “we did anything” change to “we did not do anything”

Line 635 “some sort some not so short” change to “some short some not so short”

Line 918 Grammar

Author Response

COMMENTS OF THE FIRST REVIEWER

ANSWERS OF THE AUTHORS

Line 30-31 Why does autism present a challenge in the pandemic? Would not it be better to phrase it that the pandemic presents even a more difficult challenge for people with autism and explain why.

We corrected it and we added:

“The pandemic presents even a more difficult challenge for people with autism due to high levels of anxiety, sensory issues (for example they may face difficulty wearing masks) and their inability to adjust in the changes in their daily routine [3].”

Page 2 talking about the pandemic – I would recommend using past tense

We used past tense

Line 68 “The findings of research on the difficulty of achieving balance in the family of a child with autism [21], combined with the broader picture of special education in Greece [22,23],” Could you be more specific on both points?

We corrected it and we added:

The findings of research on the difficulty of achieving balance in the family of a child with autism due to the immense pressure that child’s needs impose to the family [25]. In addition the broader picture of special education in Greece is not ideal due to the s lack of personnel and suitable buildings with all the necessary equipment and accommodations for students with ASD [26, 27

Line 121 Grammar: rephrase: “whether they were new to treatment or people who were continuing a therapeutic intervention [47].”

We changed that

Line 126 “diagnosis” change to “diagnostic process” or “assessment”

We changed it

Line 134 “typical” schools change to “mainstream”

We changed it

Line 133-135 It would be useful to specify how many children with ASD (or what percentage) attend special schools.

Unfortunately, we don’t have official percentages of children with ASD that they attend special schools (we have only percentages of students who attend special schools and they have a variety of disabilities. The majority of students with moderate or high functioning ASD attend mainstream class with the support of a specialized teacher.

Line 160-161 “accessibility of children with ASD” perhaps “accessibility of services for children with ASD”

We changed it

Line 171 “mentally ill,” rephrase “people with mental illness”

We changed that

Line 179 only educators? perhaps also other service providers

We added:

“Psychologists, logo pathologists, occupational therapists and social workers”

Line 183 use past tense

We changed that

Line 228 Table 2 Missing number in dates

We corrected it

Line 236 special educator located at home – unclear whose home? maybe special educator available online or over the phone?

We corrected it and we added:

“special educator available online or over the phone”

Line 237 “Only three families were families with one child with ASD” It is not clear if it means they had other children and only one of them had ASD, or if they only had one child. Rephrase: The child with ASD was the only child. Overall, in the participant table it is not clear if the families only had one autistic child (it looks like it), but families often have more than one autistic child.

We made that clear it is a family with only one child (that is diagnosed with ASD)

Table 3 Does not provide any new information. It is not necessary.

We deleted it

Line 278 Thematic map – I find it unnecessary. This is not a thesis. It is enough to state what the themes and subthemes were as you did in the paragraph before.

We deleted it

Line 296 Table with quotes. Again unnecessary. The quotes are presented later in the results and that is enough for this study. Also, some quotes could use checking for translation (e.g. explosion – meltdown).

We deleted it.

We change the translation.

Line 311 Thematic map is unnecessary. It just repeats the information that was already said in the text.

We deleted it

Line 489 Here the subthemes are not listed in the text, so the figure makes sense, however, I would recommend the dame approach in all themes discussed, preferably, listing them in the text. (Same for Theme 3).

We changed it and we added the subthemes in the text

Line 502 What do you mean by deepening and expansion of the condition? It seems to be suggesting that their autism is getting worse. Maybe rephrase that the situation had a more negative impact with more time spent in isolation?

We corrected it and we added:

“the situation had a more negative impact with more time spent in isolation”

Line 599 “we did anything” change to “we did not do anything”

We changed it

Line 635 “some sort some not so short” change to “some short some not so short”

We changed it

Line 918 Grammar

We corrected it

Reviewer 2 Report

General remark: The article is interesting but much too long.

Introduction:

Information on social surveys in the described periods P1 and P2 should be added. For the context of the study, the data on how Greek society approached the pandemic in these two periods is important. Has the fear diminished? Were the restrictions less obeyed?

LINES 84-86 „Previous epidemic crises  (SARS, H1N1, MERS, EBOLA) cannot be compared to the current one and this makes it difficult to systematically predict the consequences” Has society in your country been affected by these epidemics in any way?

Methods:

Currently, it is rather avoided to put forward hypotheses in qualitative research.

Is the transcription available in an open repository?

Lines. 237-238 „Only three families were families with one child with ASD” Does it mean that they have only one child, or the others have more than one child with ASD. I know the answer, because it is in Table no. 3., but in the scientific manuscripts such details are important.

Lines 252-253 „which has been recognized as one of the best research techniques in the context of social and anthropological research”. Thematic analysis is not a research technique, it is an analysis method.

Did you use any „steps” for thematic analysis e.g. Braun and Clark, 2006?

Results

General remark: I get the impression that the author/s wanted to show all the results of the study, but the articles have to "summarize" it. Usually, we show only 1-2 best quotes in the article, the important points from others are just mentioned in the description. Please think from what you can resign, to make the text shorter and easier to read.

I am sure table 4 should be moved to the attachment. Results which are later are enough.

3.2.1 In the description you should mention the difficulties with computers. „She kept losing the camera, losing the microphone, a lot of times we had problems”

3.2.6 Please mind everything is italic.

Discussion

General remark: I get confused in the discussion about what is the result of this research and what is the result of others. You have to clearly separate this.

Limitation – You should indicate limitations that your research has – CLEARLY. 1) You conducted a telephone interview – this is not the best possible way to do interviews. 2) Your interviews were short. 3) Attached “scenario” shows that a lot of questions should to be rethought so you will get more deep answers e.g. “. Did he/she expressed emotional tensions (anxiety, stress, nervousness, tantrums ) when the child had to teleconference?” can be asked „Can you tell me about any emotional tensions that your child experienced during the teleconference? 4) Small group.

Conclusion

Please propose something that can be done by government/local government/ schools/teachers to help families with children with ASD to make their life easier in the pandemic.

Author Response

COMMENTS OF THE SECOND REVIEWER

ANSWERS OF THE AUTHORS

Introduction:

Information on social surveys in the described periods P1 and P2 should be added. For the context of the study, the data on how Greek society approached the pandemic in these two periods is important. Has the fear diminished? Were the restrictions less obeyed?

We added these studies that they have been conducted in Greece and they focused on how Greek society approaches he pandemic measures:

Vatavali F, Gareiou Z, Kehagia F, Zervas E. Impact of COVID-19 on Urban Everyday Life in Greece. Perceptions, Experiences and Practices of the Active Population. Sustainability. 2020; 12(22):9410. https://doi.org/10.3390/su12229410

Anastasiou, Evgenia, and Marie-Noelle Duquenne. 2021. First-Wave COVID-19 Pandemic in Greece: The Role of Demographic, Social, and Geographical Factors in Life Satisfaction during Lockdown. Social Sciences 10: 186. https:// doi.org/10.3390/socsci10060186

Anastasiou, Evgenia, and Marie-Noelle Duquenne. 2021. What about the “Social Aspect of COVID”? Exploring the Determinants of Social Isolation on the Greek Population during the COVID-19 Lockdown. Social Sciences 10: 27. https:// doi.org/10.3390/socsci10010027

Kousi T, Mitsi L-C, Simos J. The Early Stage of COVID-19 Outbreak in Greece: A Review of the National Response and the Socioeconomic Impact. International Journal of Environmental Research and Public Health. 2021; 18(1):322. https://doi.org/10.3390/ijerph18010322

The vast majority of the Greek population complied with the restrictive measures in P1 and P2. The Greek population accepted the “stay at home” edict. The effects of these measures in the Greek population were social isolation, unemployment or changes to the employment status, and increased family conflicts.

LINES 84-86 „Previous epidemic crises  (SARS, H1N1, MERS, EBOLA) cannot be compared to the current one and this makes it difficult to systematically predict the consequences” Has society in your country been affected by these epidemics in any way?

Actually, it hasn’t affected directly our society (for example EBOLA has been restricted to Africa). It is mentioned in the Introduction as an example of how in previous epidemic crises the society had been affected and how these crises were controlled.

Methods:

Currently, it is rather avoided to put forward hypotheses in qualitative research.

We changed the word Hypothesized with this “In the current study we are investigating how the…”

Is the transcription available in an open repository?

The transcription is available to anyone that asks it

Lines. 237-238 „Only three families were families with one child with ASD” Does it mean that they have only one child, or the others have more than one child with ASD. I know the answer, because it is in Table no. 3., but in the scientific manuscripts such details are important.

We corrected it

Three of the families had only one child (the child with ASD)

Lines 252-253 „which has been recognized as one of the best research techniques in the context of social and anthropological research”. Thematic analysis is not a research technique, it is an analysis method.

We corrected it:

“Which has been recognized as one of the best analysis method in the context of social and anthropological research”

Did you use any „steps” for thematic analysis e.g. Braun and Clark, 2006?

Yes, we followed the six steps that they are proposing for thematic analysis:

1.       Familiarizing with the data.

2.        Generating initial codes

3.        Searching for themes

4.        Reviewing themes

5.        Defining and naming themes

6.       Producing the report

We added that at the analysis section

Results

General remark: I get the impression that the author/s wanted to show all the results of the study, but the articles have to "summarize" it. Usually, we show only 1-2 best quotes in the article, the important points from others are just mentioned in the description. Please think from what you can resign, to make the text shorter and easier to read.

We deleted several quotes and we focused on the most important in order to make our text easier to read.

I am sure table 4 should be moved to the attachment. Results which are later are enough.

We moved Table 4 to Appendix II

3.2.1 In the description you should mention the difficulties with computers. „She kept losing the camera, losing the microphone, a lot of times we had problems”

We added the difficulties with computers in the description.

3.2.6 Please mind everything is italic.

We corrected that

Discussion

General remark: I get confused in the discussion about what is the result of this research and what is the result of others. You have to clearly separate this.

We clarified the discussion section and we made clear our results from the findings from other researches.

Limitation – You should indicate limitations that your research has – CLEARLY. 1) You conducted a telephone interview – this is not the best possible way to do interviews. 2) Your interviews were short. 3) Attached “scenario” shows that a lot of questions should to be rethought so you will get more deep answers e.g. “. Did he/she expressed emotional tensions (anxiety, stress, nervousness, tantrums ) when the child had to teleconference?” can be asked „Can you tell me about any emotional tensions that your child experienced during the teleconference? 4) Small group.

We added clearly the limitations of our study in a new section.

“4.1. Limitations of our study

The limitations of our study are that our participants were not randomly selected and we based our research on a small group of parents. Secondly, telephone interviews may be an “easy” way to gather data but it has several disadvantages due to the lack of the face to face interaction. Finally, we could enhance our questions and include questions that could provide us with more “in deep” answers.”

Conclusion

Please propose something that can be done by government/local government/ schools/teachers to help families with children with ASD to make their life easier in the pandemic.

We added this:

“Government should develop an organized plan for an emergency situation (such as the pandemic) to support parents and their children with ASD in order to cope with the obstacles that they may arise and the high levels of stress which is emerging. Professionals should use telehealth platforms in order to evaluate the individualized needs of every child and develop the appropriate actions for crisis management. Teachers should pay attention to include parents as co-therapist to tackle the emerging situation and the immense stress, their children and themselves are facing from the pandemic.”

Round 2

Reviewer 2 Report

The changes have been well implemented.